# Efficacy of Whole Cell Inactivated *Vibrio harveyi* Vaccine against Vibriosis in a Marine Red Hybrid Tilapia (*Oreochromis niloticus* × *O. mossambicus*) Model

**DOI:** 10.3390/vaccines8040734

**Published:** 2020-12-04

**Authors:** Nadirah Abu Nor, Mohd Zamri-Saad, Ina-Salwany Md Yasin, Annas Salleh, Farina Mustaffa-Kamal, Mohd Fuad Matori, Mohd Noor Amal Azmai

**Affiliations:** 1Aquatic Animal Health and Therapeutics Laboratory, Institute of Bioscience, Universiti Putra Malaysia, UPM Serdang 43400, Malaysia; nadirahabunor1990@gmail.com (N.A.N.); salwany@upm.edu.my (I.-S.M.Y.); mnamal@upm.edu.my (M.N.A.A.); 2Department of Veterinary Laboratory Diagnosis, Faculty of Veterinary Medicine, Universiti Putra Malaysia, UPM Serdang 43400, Malaysia; annas@upm.edu.my; 3Department of Aquaculture, Faculty of Agriculture, Universiti Putra Malaysia, UPM Serdang 43400, Malaysia; 4Department of Pathology and Microbiology, Faculty of Veterinary Medicine, Universiti Putra Malaysia, UPM Serdang 43400, Malaysia; farina@upm.edu.my; 5Aquatic Animal Health Unit, Faculty of Veterinary Medicine, Universiti Putra Malaysia, UPM Serdang 43400, Selangor, Malaysia; fuma@upm.edu.my; 6Department of Biology, Faculty of Science, Universiti Putra Malaysia, UPM Serdang 43400, Selangor, Malaysia

**Keywords:** marine red hybrid tilapia, *Vibrio harveyi*, immune response, formalin-killed vaccine

## Abstract

*Vibrio harveyi* causes vibriosis in various commercial marine fish species. The infection leads to significant economic losses for aquaculture farms, and vaccination is an alternative approach for the prevention and control of fish diseases for aquaculture sustainability. This study describes the use of formalin-killed *Vibrio harveyi* (FKVh) strain Vh1 as a vaccine candidate to stimulate innate and adaptive immunities against vibriosis in a marine red hybrid tilapia model. Tilapia are fast growing; cheap; resistant to diseases; and tolerant to adverse environmental conditions of fresh water, brackish water, and marine water and because of these advantages, marine red hybrid tilapia is a suitable candidate as a model to study fish diseases and vaccinations against vibriosis. A total of 180 healthy red hybrid tilapias were gradually adapted to the marine environment before being divided into two groups, with 90 fish in each group and were kept in triplicate with 30 fish per tank. Group 1 was vaccinated intraperitoneally with 100 µL of FKVh on week 0, and a booster dose was similarly administered on week 2. Group 2 was similarly injected with PBS. Skin mucus, serum, and gut lavage were collected weekly for enzyme-linked immunosorbent assay (ELISA) and a lysozyme activity assay from a total of 30 fish of each group. On week 4, the remaining 60 fish of Groups 1 and 2 were challenged with 10^8^ cfu/fish of live *Vibrio harveyi*. The clinical signs were monitored while the survival rate was recorded for 48 h post-challenge. Vaccination with FKVh resulted in a significantly (*p* < 0.05) higher rate of survival (87%) compared to the control (20%). The IgM antibody titer and lysozyme activities of Group 1 were significantly (*p* < 0.05) higher than the unvaccinated Groups 2 in most weeks throughout the experiment. Therefore, the intraperitoneal exposure of marine red hybrid tilapia to killed *V. harveyi* enhanced the resistance and antibody response of the fish against vibriosis.

## 1. Introduction

Vibriosis is a disease that is caused by *Vibrio* spp. and is a major disease of commercial marine fish [1]). The first confirmed vibriosis in fish was reported by [2], when an epizootic among migrating eels, *Anguilla vulgaris* caused by a bacterium called *Bacillus anguillarum*, was reported in 1817. Later in 1909, the bacterium that was responsible for disease outbreaks in *A. anguilla* in Sweden was isolated and identified as *V. anguillarum* [3]. In subsequent years, more outbreaks associated with *Vibrio* spp. infections among wild fish such as saithe and cod were reported [4]. However, this disease started to catch serious attention when it became a threat to farmed fish, especially in the North America, Europe, and Japan. In Asia, *Vibrio* spp. was first reported to affect the yellowtail *Seriola quinqueradiata* culture industry in Japan in 1963 [5]. Vibriosis also caused losses in cultured coho salmon and seabream in Japan [6].

Vaccination is an alternative method of controlling fish diseases apart from the use of antibiotics [7]. The purpose of vaccination is to induce and build resistance in the host against a specific pathogen [8], and injectable vaccines are preferred because the antigen dose is known, thereby making it easy to correlate the antigen dose with vaccine protection [9]. Besides this, the injection method is the most potent route of vaccination, as it produces a stronger immune response compared to other routes of vaccination [10]. The extracellular and surface proteins of bacterial pathogens have the advantage of easy recognition by the infected host and, thus, are more likely to serve as targets for vaccine development. Extracellular or surface proteins of pathogens have been found to possess vaccine potential and confer immune protection [11].

However, the testing of vaccines against vibriosis in real hosts such as Asian seabass (*Lates calcarifer*), grouper (*Epinephelus* spp.), and snapper (*Lutjanus* spp.) is expensive [12,13] and not easy to conduct, particularly on the laboratory scale. Thus, finding a suitable animal model to study diseases of marine fish such as vibriosis is important and crucial. Tilapia (*Oreochromis* spp.) culture has surged to become one of the leading and cheap farmed fish species throughout the world. Tilapia can tolerate a wide range of salinities and is considered to be resistant to diseases [14]. Additionally, they have a high survival rate and fast growth. Since tilapia are well-studied, fast-growing, cheap, easy to culture, and widely cultured in the world, it is worth exploring the potential use of tilapia as an alternative animal model to study formalin-killed *Vibrio* vaccine towards immune responses against vibriosis.

## 2. Materials and Methods

### 2.1. Experimental Fish

A total of 180 clinically healthy red hybrid tilapia (*Oreochromis niloticus* × *O. mossambicus*) weighing 100 ± 10 g with lengths of 4–5 inch were selected for this study. They were divided into two groups: Groups 1 and 2 with triplicate where 30 fishes per tank and 90 fishes per group. After two weeks of acclimatization in fresh water, the fish were gradually introduced into sea water with 2 ppt increments per day until they reached 15ppt. The tanks were filled with pre-treated (sand filter and UV light) seawater gradually with adequate aeration, and one third of the water was exchanged daily. During acclimatization, the light cycle was held constant with 12 h of lighting per day. Throughout the experimental period, the water temperature, pH, salinity, and dissolved oxygen were maintained at 27.5 ± 0.5 °C, 7.8 ± 0.2, 25 ± 0.1 ppt, and 5.0 ± 0.4 mg/L, respectively. The fish were fed twice a day with a commercial tilapia feed at the rate of 10% body weight. All the experiments on fish were approved by the Institutional Animal Care and Use Committee, Universiti Putra Malaysia.

### 2.2. Bacterial Strains and Culture

The pathogenic *Vibrio harveyi* strain Vh1 was used in this study. The bacterium was previously isolated from diseased farmed grouper (*Epinephelus* sp.) in Malaysia [13]. The identification of this strain was carried out using 16S rRNA analysis. The strain was maintained in thiosulphate-citrate-bile-salts-sucrose (TCBS) agar (Oxoid, Hampshire, UK) and tryptone soya broth/agar (TSB/TSA) (Oxoid) with the addition of 1.5% (*w*/*v*) NaCl in glycerol stock at −80 °C for long-term storage. Prior to the start of the experiment, the virulence of the bacterium was revived using the protocol of Koch’s postulate, which was carried out twice in red hybrid tilapia.

The stock culture of the *Vibrio harveyi* strain Vh1 was grown in TSB supplemented with 1.5% NaCl and incubated in an incubator shaker (Daiki Science, Seoul, Korea) at 200 rpm at 30 °C for 24 h. Following incubation, the broth containing the bacteria was centrifuged at 12,000 rpm for 15 min at 28 °C. The bacterial concentration in the solution was determined by the serial dilution and colony counting methods.

### 2.3. Preparation of Formalin-Killed Vibrio Harveyi (FKVh) Vaccine

Once the *V. harveyi* strain Vh1 was subcultured from the stock and the concentration was determined, the suspension was adjusted to a concentration of 10^9^ CFU/mL. The bacteria were then killed by introducing buffered formalin to the end concentration of 0.5% formalin in phosphate buffered saline (PBS) and kept overnight at 4 °C. Then, the bacterial cells were subsequently washed with PBS by centrifugation at 12,000 rpm and re-suspended. The next day, the suspension was tested for complete inactivation by plating onto TSA with 5% goat blood (Oxoid). No growth confirmed complete inactivation [15].

### 2.4. Vaccine Safety Test

To test the safety of the vaccine, 10 fish were injected intraperitoneally with 1 mL of the vaccine at a concentration of 10^9^ cfu/fish. Then, they were observed for 14 days for adverse side effects, which included behavior changes such as lethargy, loss of appetite, aggression, isolating, color change, mortality, and other clinical signs related to IP injection at a high dose [16]. A safe vaccine would not produce these changes. In this study, the dosage of vaccination was 10^6^ cfu/fish, while the dose used for the vaccine safety test was 10^9^ cfu/fish.

### 2.5. Experimental Design

At the start of the experiment, 180 healthy fish were divided into two groups, with 90 fish in each group with triplicates of 30 fishes per tank. Group 1 was vaccinated intraperitoneally with 100 µL of the FKVh vaccine containing 10^6^ cells/fish on week 0. The booster dose was similarly administered on week 2. Group 2 was similarly injected with 100 μL of PBS on weeks 0 and 2. Prime vaccination was the immunization strategy involving introducing the host to a certain immunogen specific to certain vaccine while booster was given after the first vaccination to boost and prolong the protection against disease, as after certain period of time the antibody level of fish will decrease. [10] proved in his study and stated that the administration of second booster dose provides a longer period of protection that lasts for at least 12 weeks at a rate of 70% protection and later challenged via intraperitoneal with 100 μL of the inoculum containing 10^8^ CFU/fish (LD80) of live virulent *V. harveyi* on week 4 [17]. Clinical signs and mortality rate were recorded hourly for 48 h after the challenge, and the survivors were sacrificed at the end of week 10 of the experiment (Figure 1).

Samples of skin mucus, serum, and gut lavage were collected from 3 fish of each group at a weekly interval throughout the 10-week study period for enzyme-linked immunosorbent assay (ELISA) and lysozyme activity assay.

A total of 60 fish from each group were challenged via intraperitoneal injection of 100 μL of the inoculum containing 10^8^ CFU/fish of live virulent *V. harveyi* on week 4 [17]. Clinical signs and mortality rate were recorded hourly for 48 h after the challenge, and the survivors were sacrificed at the end of week 10 of the experiment (Figure 1). The gross lesions of both dead and sacrificed fish were also recorded before samples of organs such as liver, kidney, brain, spleen, and gill were collected and processed for histopathology assessments. The protection efficacy of the vaccine was determined by comparing the percentage of mortality of the vaccinated and control groups.

#### 2.5.1. Mucus

The fish were anesthetized using MS222 at the concentration of 120 mg/L before a mucus sample was collected by swabbing the skin 10 times from the most anterior part of the head to end of caudal fin on one side of each fish using sterile cotton bud. The swabs were then put into test tubes that contained 3 mL of PBS, supplemented with sodium azide at 0.02% (*w*/*v*) [17]. Then, the suspensions were stored overnight in chiller at 4 °C. The next day, the suspension was vigorously vortexed for 2 min and the solution was separated by centrifugation at 3000× *g* for 10 min, then transferred into 1.5 mL micro-centrifuge tubes (Eppendorf, Hamburg, Germany) and kept at 4 °C before being stored at −20 °C for the measurement of the antibody titer using ELISA and lysozyme activity assay.

#### 2.5.2. Serum

Following anesthesia, the fish were bled from the caudal vein using 25 G × 1 needles (Terumo, Tokyo, Japan) according to [17] and 1 mL blood was collected into a 3 mL syringe (Terumo). The blood samples were then transferred into 1.5 mL micro-centrifuge tubes and were held at 25 °C for 1 h before the serum was separated by centrifugation at 3000× *g* for 10 min. The serum was then kept at 4 °C before later stored at −20 °C for the measurement of the antibody titer using ELISA and lysozyme activity assay.

#### 2.5.3. Gut Lavage

Gut lavage was obtained by cutting the gut, approximately 10–15 cm in length, posterior to the stomach before infusing 1 mL of sterile PBS supplemented with 0.02% (*w*/*v*) sodium azide into the gut lumen. The gut was gently massaged before the fluid was collected into a 1.5 mL micro-centrifuge tube and centrifuged at 3000× *g* for 10 min [17]. The supernatant was then collected and kept at 4 °C before being stored at −20 °C for the measurement of the antibody titer using ELISA and lysozyme activity assay.

### 2.6. Sample Processing

#### 2.6.1. Enzyme-Linked Immunosorbent Assay (ELISA)

Serum, gut lavage and body mucus samples were subjected to indirect ELISA using the method described by [18] with minor modifications. Two colonies of *V. harveyi* from TCBS agar were sub-cultured into 100 mL of the TSB supplemented with 1.5% NaCl and incubated in incubator shaker (Daiki Science, Seoul, Korea) at 200 rpm at 30 °C for 24 h. The final concentration was determined at 4.5 × 10^6^ CFU/mL. The bacterial suspension was washed with PBS three times to remove the remaining TSB. In between each washing, the suspension was centrifuged at 5000 rpm for 15 min. The pellet was then re-suspended in coating buffer (pH 9.6) and was boiled in water bath for 20 min, left to cool at room temperature before being used in the ELISA procedure.

To perform the ELISA, a microtiter plate was coated with 100 µL in triplicate of the whole cells of *V. harveyi* suspension prepared earlier. The plates were then left overnight at 4 °C before being washed twice with PBS containing 0.05% Tween 20 (PBST). The reaction was blocked with 200 µL of blocking buffer of PBS with 1% BSA (PBS/BSA) before the plates were washed twice. Then, 100 µL of mucus, serum and gut lavage using same dilution of 1:1000 was added and incubated for 1 h at 37 °C. Following incubation, the plates were washing with PBST and 100 µL per well of goat anti-tilapia immunoglobulin serum diluted at 1:5000 was added into each well and incubated for another 1 h. After washing three times with PBST, 100 µL of conjugated rabbit anti-goat IgM-horseradish peroxidase (Nordic, Susteren, Netherland) diluted at 1:5000 was added into each well and incubated again for another 1 h. After being washed three times with PBST, 100 µL of substrate containing tetramethyl- 3.3′,5,5′-benzidine (Merck, Kenilworth, NJ, USA), dimethyl sulfoxide (DMSO) (Merck), 0.1 M sodium acetate/citric acid buffer, and 31% hydrogen peroxide was added into each well and further incubated for 30 min. The reaction was stopped by adding 50 µL 2.5 M sulfuric acid per well. All the the plates then were read at 450 nm of wavelength (340st: Anthos Zenyth, Salzbury, Austria).

#### 2.6.2. Lysozyme Assay

Lysozyme activity was determined according to the method of [19] based on the lysis of the lysozyme sensitive Gram-positive bacterium, *Micrococcus lysodiekticus* (Sigma-Aldrich, St Louis, MO, USA), with slight modifications. A total of 25 µL of the serum, mucus, and gut lavage fluid was added into 75 µL of lyophilized *M. lysodeikticus* cell suspension (Sigma, 75 mg/mL) prepared with 0.1 M phosphate citrate buffer and pH 5.8 in wells of a 96-well plate in triplicate. After rapid mixing, the change in turbidity was measured every 30 s for 5 min at 450 nm. The absorbance was measured continuously for 1 h at 450 nm. A unit of lysozyme activity was defined as the amount of enzyme causing a decrease in absorbance of 0.001 per min and expressed as U/mg unit.

#### 2.6.3. Histopathology Analysis

Histological analysis (Figure 5) was conducted to investigate the pathological changes following the challenge of vaccinated and non-vaccinated red hybrid tilapia. Organs samples were collected from dead fish and were processed for histological examination. Samples were dehydrated for 2 h in a series of alcohol solutions starting with 50%, 70%, and 90%, followed by a clearing process in xylene. The samples were then impregnated with paraffin and melting point at 56–58 °C, followed by embedding in melted paraffin while ensuring the correct orientation of tissues, and sectioned at 4 µm for slide preparation. Serial 4–5 µm thick sections were cut from paraffin blocks onto the glass slides. The sections were allowed to dry overnight at 40 °C. All the samples were processed and subjected to haematoxylin and eosin (H&E) staining [17]. The sectioned tissues were observed under light microscope FIVE Image Analyzer (Olympus, Kuala Lumpur, Malaysia) at 100× and 400× magnifications for histological changes.

### 2.7. Data Analysis

Antibody level and lysozyme activity in serum, mucus, and gut lavage fluid were compared using one-way analysis of variance (ANOVA) following post hoc Tukey’s test. All the statistical analyses were performed via IBM SPSS Statistics^®^ version 22.0 (Armonk, NY, USA: IBM Corp).The values were considered significantly different at *p* < 0.05.

### 2.8. Ethics Statement

The application of *V. harveyi* strain Vh1 and its parental pathogenic strain conformed to the guidelines stipulated by the Department of Biosafety, Ministry of Natural Resources and Environment, Malaysia, under approval number JBK(S) 602-1/2/136(6). All the animal experiments were approved by the Institutional Animal Care and Use Committee, Universiti Putra Malaysia.

## 3. Results

### 3.1. Antibody Response in Mucus, Serum and Gut Lavage

The IgM antibody levels in the skin mucus are presented in Figure 2a. The antibody levels were consistently and significantly (*p* < 0.05) higher in the vaccinated Group 1 compared to the unvaccinated Group 2 for the first 8 weeks of the study period. The administration of the booster dose on week 2 enhanced the antibody response further and remained significantly (*p* < 0.05) high until week 8. The antibody level in the vaccinated Group 1 reached a peak at week 4, before starting to decline and becoming insignificant (*p* > 0.05) at weeks 9 and 10.

The serum IgM antibody levels are summarized in Figure 2b. Immunization with FKVh resulted in significantly (*p* < 0.05) higher IgM antibody levels in the serum of Group 1 compared with the unvaccinated Group 2 throughout the study period. The antibody levels in vaccinated Group 1 increased significantly (*p* < 0.05) following vaccinations at weeks 0 and 2 but decreased at week 4.

The antibody levels in the gut lavage fluid are shown in Figure 2c. The vaccinated Group 1 showed significantly (*p* < 0.05) higher antibody levels throughout the 10-week study period when compared with the non-vaccinated Group 2. The booster dose on week 2 enhanced the gut lavage antibody levels to reach a peak on week 8 and remained significantly (*p* < 0.05) high until week 10.

### 3.2. Lysozyme Activity in Body Mucus, Serum and Gut Lavage Fluid

The activities of lysozyme in the body mucus of marine red hybrid tilapia of Groups 1 and 2 are summarized in Figure 3a. The activity was detected as early as week 1 in the vaccinated Group 1 after the first vaccination was given on week 0. However, the booster dose on week 2 slowly decreased the lysozyme activities in week 3 but kept showing an increment pattern until week 10, while Group 2 showed a lower lysozyme activity in most of the weeks compared to Group 1.

The activities of lysozyme in the serum of Groups 1 and 2 are summarized in Figure 3b. The lysozyme activity in Group 1 was detected to be significantly (*p* < 0.05) higher compared to Group 2 as early as week 1, but slowly declined by week 3. The highest activity in serum of Group 1 was in week 3, with 283 ± 2.58 U/mg.

The lysozyme activities in the gut lavage fluid are summarized in Figure 3c. The activity of lysozyme in the gut lavage of Group 1 was detected as early as week 1, and the activity was the highest after the first vaccination was given on week 0 and remained significantly (*p* < 0.05) higher compared to Group 2 throughout the 10-week study period.

### 3.3. Relative Percentage Survival and Clinical Signs

Virulent *V. harveyi* was able to infect and kill the marine red hybrid tilapia as early as 4 h post-challenge (Figure 4). These findings were similar to those of [20] and [10] as using high dose challenge; 10^9^ cfu/mL, the mortality of fish was observed as early as 24 h post-challenge in unvaccinated group. Overall, cumulative mortality of 80% was recorded in unvaccinated Group 2. Within the first 12 h post-challenge, mortality involved 34 fishes (57%) and by 48 h, 48 fishes (80%) died in Group 2. On the other hand, overall rate of mortality in Group 1 was 13%. The surviving fish of Group 2 showed mild lesions at the site of abrasion, while the surviving fish of Group 1 showed no clinical sign.

### 3.4. Pathological Changes in Organs

Following challenge, there were hemorrhages in the gills, particularly the connective tissue with increments of the Goblet cells. There were detachments and fusions of the epithelium of secondary lamella which were absent in Group 1. Severe congestion was seen in the brain of Group 2 m while Group 1 showed mild congestion (Figure 5).

## 4. Discussion

Bacterial infections in fish, especially infections with Gram-negative bacteria, are still a foremost problem for the fish industry [21]. *Vibrio* spp. are Gram-negative bacteria commonly isolated from different ecosystems as well as aquaculture farms [22]). Many studies have shown that various species of *Vibrio* could lead to a systemic bacterial infection known as vibriosis. The disease is widely accused of causing a high mortality amongst fish in marine aquaculture systems worldwide [23,24]. Marine tilapia or tilapia that have been adapted to a marine environment could be a cheaper and excellent model by which to study the disease development and prevention of vibriosis of marine fish.

This study describes the systemic and mucosal immunities and protective capacity of FKVh vaccine against vibriosis in a marine red hybrid tilapia model. Therapeutic measures are generally ineffective, thus the development and study of vaccine in a potential fish model is essential to control the bacterial infection, as the overuse of antibiotics against vibriosis might increase the number and types of drug-resistant *Vibrio* spp. [24,25,26]. Formalin-killed vaccine, a whole cell inactivated vaccine, was chosen in this study since previous studies have concluded that there was insignificant difference between adverse types of vaccine in term of mortality and the rate of survival [27,28].

The intraperitoneal route was chosen as it is well documented that this injection method triggers a systemic immune response with high antibody levels in a short period of time. This method provides longer protection compared to oral and immersion, which trigger local mucosal immunity [10]. This might have a large impact on the human health and the environment [29].

Our results indicate that fish vaccinated with the FKVh vaccine produce higher antibody levels in serum, mucus, and gut lavage compared to non-vaccinated fishes as early as week 1. This vaccination has a strong promoting effect in inducing both humoral and cellular immune responses in fish [30]. Due to significant high mucosal and systemic antibody responses, the protection conferred by this vaccine was evaluated by challenging the fish with 10^9^ CFU/mL of live *V. harveyi*. The FKVh vaccination resulted in higher survival rate, indicating higher protection in vaccinated group, believed to be due to protective and prolong antibody production either in the serum or at mucosal level.

Lysozyme is made up from a single polypeptide chain of about 120 amino acids with a molecular weight of around 14.4 kDa. Lysozyme or muramidase provides non-specific defence to the fish by splitting the linkages between N-acetylmuramic acid and N-acetylglucosamine in the peptidoglycan layers of both Gram-positive and Gram-negative bacteria, leading to cell death [31]. The optimum activity for lysozyme occurs between pH 5 and 7 [32]. Nevertheless, there are several factors that influence the activity of lysozyme, which include sex, age and size, season, water temperature, pH, toxicants, infections, and stress [31]. These explained the higher activity of lysozyme in vaccinated group, especially after being challenged. The present study revealed that lysozyme activities could be detected in the body mucus, serum, and gut lavage as early as 7 days post-vaccination. This important immune substance is crucial for the primary immune defence of marine red hybrid tilapia to ensure their survival. Furthermore, this enzyme was reported to occur in lymphoid tissue, blood plasma, and other body fluids of either freshwater or marine fish and plays important roles in fish innate immunity, especially in juveniles due to the high surface to volume ratio in the early stage of life, which declines with growth [33]. Ref. [34], the Goblet’s cells that secrete mucus containing lysozymes and other innate immune substances started to develop between days 15 and 20 in Senegal sole (*Solea senegalensis*) fingerlings. This explains the existence of lysozyme activities in marine red hybrid tilapia with a size of 100 ± 10 g.

Subsequently, the percentage mortality of the non-vaccinated fish following the challenge with live *V. harveyi* was higher than in the vaccinated fish. This is because of the induction of mucosal immune responses [35] and crucial long-lasting immune protection [36]. A high dose concentration of live bacteria for challenge was choose according to [20] and [10], who proved that high dose challenge produced clinical signs and mortality as early as 24 h post-challenge.

Vaccination enhanced the resistance of fish against challenge, resulting in relatively low rate of mortality. In fact, [37] have concluded that the vaccination of fish with formalin-killed whole cells resulted in an approximately 80% survival, which is quite similar to the findings of this study. This is because antibodies have the ability to directly neutralize pathogens by binding to the bacteria or by interfering with the receptors used by the bacteria to infect cells [38]. In addition, the innate immune system, such as the lysosome activity, is the first line of host defence in dealing with pathogens until the adaptive immune system is able and potent enough to take over [39].

The analysis of histological changes is important in the diagnosis of diseases, besides providing information on the mechanism of pathogens. The histological changes caused by a pathogen may vary, showing that the behavior of the infection depends not only on the pathogen but also on the health status of the host [40]. Fish with a better nutritional and immune status are more able to control infection and have a lower severity in terms of tissue damage. For the histology analyses in this study, tissues from non-vaccinated fish showed moderate to severe lesions compared to the vaccination group.

## 5. Conclusions

In conclusion, formalin-killed *V. harveyi* strain Vh1 provided a better protection against vibriosis in a marine red hybrid tilapia model by producing higher antibody responses and more efficient innate defence. Therefore, this FKVh vaccine is a suitable vaccine candidate against vibriosis. Marine red hybrid tilapia is also a suitable animal model to study bacterial fish diseases, as it can respond to vibriosis, which is a common disease of commercial marine and brackish water fish. In addition, further research on suitable adjuvants is also needed in order to develop an injection vaccine. Furthermore, the use of other kinds of adjuvants or nanoparticles to formulate the vaccine is also a concern for improving the efficacy of an inactivated whole-cell FKVh vaccine.

## Figures and Tables

**Figure 1 vaccines-08-00734-f001:**
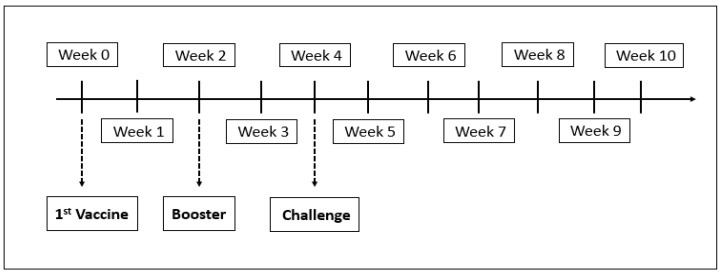
Timeline of the vaccination regime and challenge assay for Group 1 and Group 2.

**Figure 2 vaccines-08-00734-f002:**
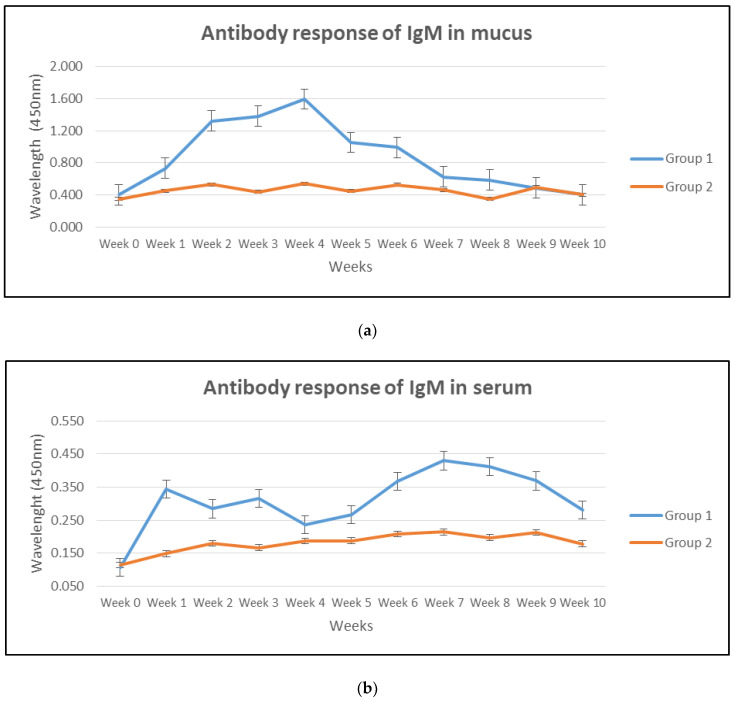
Specific (**a**) mucus, (**b**) serum, and (**c**) gut lavage IgM antibody titers were determined using ELISA assay in Group 1 and 2. First vaccination and booster were given on week 1 and 2, while the challenge was on week 4. Error bar indicates significant differences at *p* < 0.05.

**Figure 3 vaccines-08-00734-f003:**
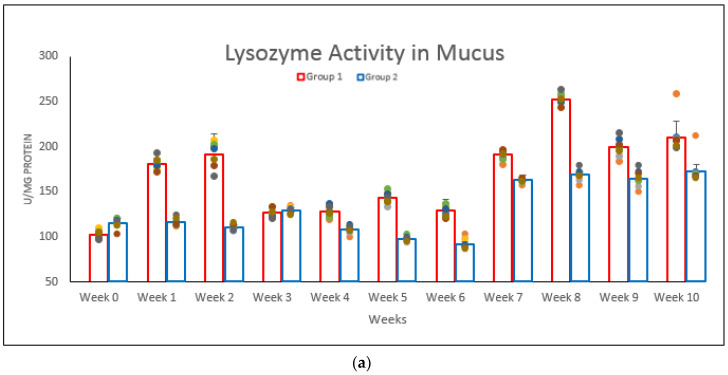
Lysozyme activity: (**a**) mucus, (**b**) serum, and (**c**) gut lavage of marine red hybrid tilapia following vaccination with formalin-killed *V. harveyi* for both Group 1 and 2. Relatively, the lysozyme activity was plotted directly with the value of the optical density, which was inversely converted. Error bar indicates significant differences at *p* < 0.05. The different dot colors from each bar indicated several readings each weeks from each groups.

**Figure 4 vaccines-08-00734-f004:**
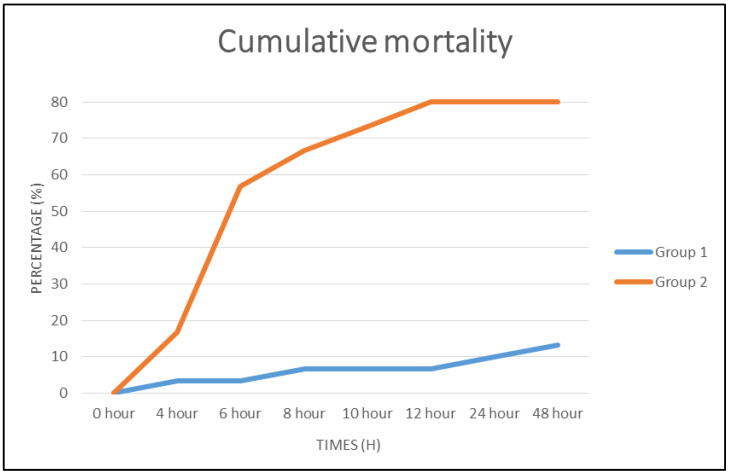
The cumulative mortality of marine red hybrid tilapia post-challenge with 10^8^ CFU/mL of live virulent *Vibrio harveyi* by intraperitoneal injection. Group 1 = vaccinated; Group 2 = non vaccinated.

**Figure 5 vaccines-08-00734-f005:**
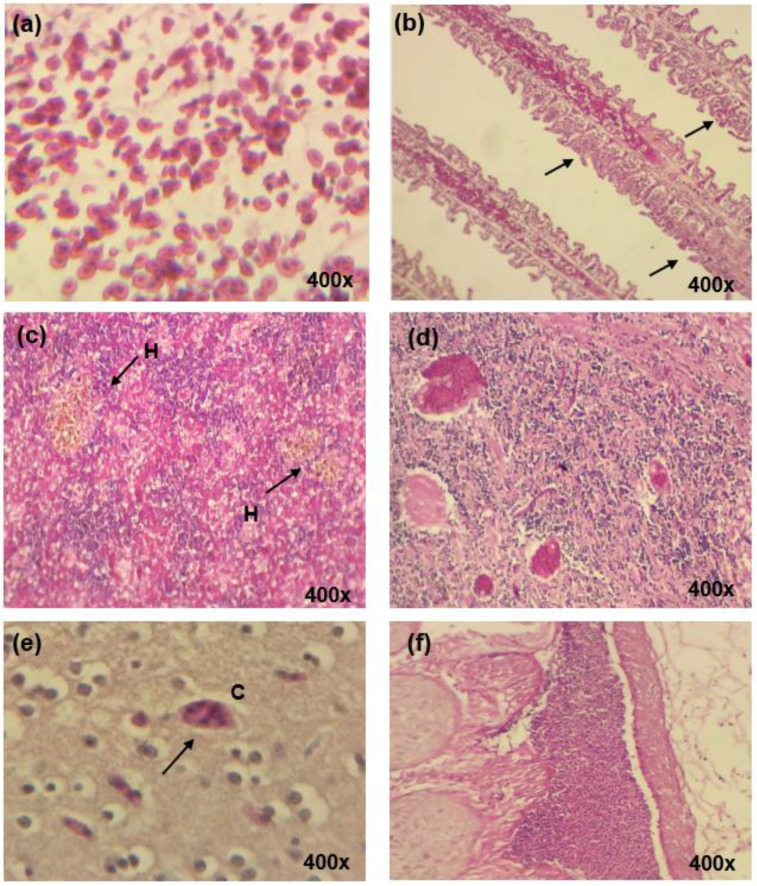
Histological changes in the organs of marine red hybrid tilapia after being challenged with *Vibrio harveyi.* (**a**) Gill from non-vaccinated fish showed hemorrhages at the connective tissue (H&E × 400); (**b**) fusion (black arrows) and detachment of secondary lamellae in non-vaccinated fish (H&E 400); (**c**) presentation of hemosiderin-phages (H) in the spleen tissue of non-vaccinated fish (H&E × 400); (**d**) brain tissue showing severe congestion in non-vaccinated fish (H&E × 400); (**e**) brain tissue from vaccinated fish showing mild congestion (C) (black arrows) (H&E × 400); (**f**) gill from a vaccinated fish showing the presence of gill-associated lymphoid tissue (GIALT) (H&E × 400).

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
