# Peer review of "Efficacy of Whole Cell Inactivated Vibrio harveyi Vaccine against Vibriosis in a Marine Red Hybrid Tilapia (Oreochromis niloticus × O. mossambicus) Model"

_vaccines, 2020, doi:10.3390/vaccines8040734_

Round 1

Reviewer 1 Report

The present study address the efficacy of a vaccine based on formalin-inactivated Vibrio harveyi against vibriosis in Tilapia. The results in terms of survival advantage provided by the vaccine are impressive. However, many of the experimental descriptions need clarification. Whether prime and boost is required should be addressed in combination with adjuvants, in order to know whether providing adjuvanticity allows to reduce the number or required vaccinations.

MAJOR CONCERNS

1.- The experimental design described in the abstract does not match the one described in the Material and Methods section. In the abstract the scheme would be vaccination and infection after 14 days. However, according to the “Experimental design” and Figure 1, two doses of vaccine were administered at weeks, 2 and 4, what represents a prime-boost scheme. If so, please, specify it in the main text.

2.- Assuming that the scheme depicted in Figure 1 is correct (then, please, clarify the text in the abstract), a comparison between prime, boost and prime-boost would be needed in order to understand the need of two vaccine administration.

3.- In this line, this quote indicates the use of formalin-fixed bacteria plus CpG as adjuvant. The current study does not include adjuvant in the vaccine formulation. It would be interesting to see the effect of it under prime-only scheme, in order to know whether it allows to reduce the number of vaccine administrations.

4.- Only one dose of live bacteria was used, and data clearly reveal that it is a quite high dose capable of killing up to 80% of non-vaccinated individuals. This result is, in terms of survival is pretty impressive and illustrative of the effectiveness of the vaccine. However, the remaining population after vaccination is very different between vaccinated (85%) and non-vaccinated (20%) groups. This could bias the interpretation of the results when analyzing long-term parameters. In order to solidify the mechanisms responsible for the protection, a low-dose challenge should be performed with no such a big difference in survival between groups.

5.- Authors indicate that formalin-killed vaccine form is primarily considered. However, they only provide one reference. It would be interesting to test alternative (heat-killed?) in order to compare the effectiveness between them.

MINOR CONCERNS

6.- Abstract indicates that only Group 1 was challenge with live Vibrio harveyi. I believe this is not correct according to the main text. Also, abstract indicates that this group was challenged with 10^9 cfu/mL of live bacteria. According to the “Experimental design”, fishes were infected with 100 uL of this solution, meaning 10^8 total bacteria. This is the real amount of pathogen used for infection and what should be indicated rather than the concentration without specifying the actual amount of pathogen.

7.- How can the authors know the final concentration of the vaccine? For collecting the bacteria previous to their killing, they centrifuge the culture at 12.000 rpm. However, after fixation, the washing were performed by centrifugation at 2000 rpm. What is the reason for this change? Are fixed bacteria pelleted at this low speed?.

8.- The vaccine safety test were performed by injecting 1mL of the vaccine (I guess that at 10^9 CFUs), therefore injecting 10^9 FKVh). Was this so high dose performed on purpose?

9.- In order to make informative the histological images, please, show pictures from both vaccinated and non-vaccinated individual to compare the histological changes.

Author Response

Revised version: 10 November 2020

Submission ID: vaccines-985551

Title: EFFICACY OF WHOLE CELL INACTIVATED VIBRIO HARVEYI VACCINE AGAINST VIBRIOSIS IN MARINE RED HYBRID TILAPIA (Oreochromis niloticus × O. mossambicus) MODEL

No.

Reviewer comments/ suggestions

Authors responds

Reviewer 1

1

The present study addresses the efficacy of a vaccine based on formalin-inactivated Vibrio harveyi against vibriosis in Tilapia. The results in terms of survival advantage provided by the vaccine are impressive. However, many of the experimental descriptions need clarification. Whether prime and boost is required should be addressed in combination with adjuvants, in order to know whether providing adjuvanticity allows to reduce the number or required vaccinations.

Thank you.

2

The experimental design described in the abstract does not match the one described in the Material and Methods section. In the abstract the scheme would be vaccination and infection after 14 days. However, according to the “Experimental design” and Figure 1, two doses of vaccine were administered at weeks, 2 and 4, what represents a prime-boost scheme. If so, please, specify it in the main text.

The authors already improved the vaccination schedule both in abstract and experimental design

“A total of 180 healthy red hybrid tilapias were gradually adapted to the marine environment before being divided into two groups with 90 fish in each group and were kept in triplicate with 30 fish per tank. Group 1 was vaccinated intraperitoneal with 100 uL of FKVh on week 0 and booster dose was similarly administered on week 2. Group 2 was similarly injected with PBS. Skin mucus, serum and gut lavage were collected weekly for enzyme-linked immunosorbent assay (ELISA) and lysozyme activity assay from a total of 30 fish of each group. On week 4, the remaining 60 fish of each of Groups 1 and 2 were challenged with 108 cfu/fish of live Vibrio harveyi.”

3

Assuming that the scheme depicted in Figure 1 is correct (then, please, clarify the text in the abstract), a comparison between prime, boost and prime-boost would be needed in order to understand the need of two vaccine administration.

Thank you. The authors already improved and included the explanation in experimental design as suggested by Reviewer 1. Please refer to the main manuscript in Line 116.

 “Prime vaccination was the immunization strategy by introducing host with certain immunogen. Booster was given after first-vaccination to boost and prolong the protection against disease as in certain period of time, antibody level of fish will decrease. Ismail et al. (2016) proved in his study and stated that administration of second booster dose provides longer period of protection that last for at least 12 weeks at a rate of 70 % protection.”

4

In this line, this quote indicates the use of formalin-fixed bacteria plus CpG as adjuvant. The current study does not include adjuvant in the vaccine formulation. It would be interesting to see the effect of it under prime-only scheme, in order to know whether it allows to reduce the number of vaccine administrations.

The main objectives in this study is to know whether marine red hybrid tilapia is a suitable candidate as a fish model in fish disease and fish vaccination study especially in vibriosis by studying the immune response and histopathological changes after live bacteria was introduced.

However, the author agree that further study need to be done in order to test for different vaccine efficacy by using vaccine with adjuvant or without adjuvant and also test using different types of vaccine and this suggestion already included in the conclusion.

“In addition, further research on suitable adjuvants is also needed to develop an injection vaccine. Furthermore, the use of other kinds of adjuvants or nanoparticles to formulate the vaccine is also a concern for improving the efficacy of an inactivated whole-cell FKVh vaccine.”

5

Only one dose of live bacteria was used, and data clearly reveal that it is a quite high dose capable of killing up to 80% of non-vaccinated individuals. This result is, in terms of survival is pretty impressive and illustrative of the effectiveness of the vaccine. However, the remaining population after vaccination is very different between vaccinated (85%) and non-vaccinated (20%) groups.

This could bias the interpretation of the results when analyzing long-term parameters. In order to solidify the mechanisms responsible for the protection, a low-dose challenge should be performed with no such a big difference in survival between groups.

Bacterial CFU/ml using in this study was 108 cfu/fish. This concentration was chosen after preliminary study of median lethal dose LD50.

Besides, previous study also using high dose in challenge trial. For example from Firdaus-Nawi et al., 2014 entitled “Efficacy of feed-based adjuvant vaccine against Streptococcus agalactiae in Oreochromis spp. in Malaysia “;

Ismail et al., 2016 entitled “Feed-based vaccination regime against streptococcosis in red tilapia, Oreochromis niloticus x Oreochromis mossambicus

Discussion in Line 377

“High dose concentration of live bacteria for challenge was choosen according to Firdaus-Nawi et al. (2014) and Ismail et al. (2016) who proved that high dose challenge produced clinical sign and mortality as early as 24 h post-challenge.”

6

Authors indicate that formalin-killed vaccine form is primarily considered. However, they only provide one reference. It would be interesting to test alternative (heat-killed?) in order to compare the effectiveness between them.

Thank you. Both heat-killed and formalin-killed vaccine are whole cell inactivated vaccine. The author decided to choose formalin-killed vaccine because from previous study that stated that there was insignificant difference between both types of vaccine in mortality and survival rates’s results and some study showed not much differences between the effectiveness between both. But yes, the author agree that further study in comparison the effectiveness of heat-killed, formalin-killed and autoclave-killed vaccine need to be done.

Bactol, I. D. C., Padilla, L. V., & Hilario, A. L. (2018). Immune response of tilapia (Oreochromis niloticus) after vaccination with autoclavekilled, heat-killed, and formalin-killed whole cell Aeromonas hydrophila vaccines as possible serotype-independent vaccines. International Journal of Agriculture and Biology20(4), 846-850.

Dehghani, S., Akhlaghi, M., & Dehghani, M. (2012). Efficacy of formalin-killed, heat-killed and lipopolysaccharide vaccines against motile aeromonads infection in rainbow trout (Oncorhynchus mykiss). Glob Vet9, 409-415.

7

Abstract indicates that only Group 1 was challenge with live Vibrio harveyi. I believe this is not correct according to the main text. Also, abstract indicates that this group was challenged with 10^9 cfu/mL of live bacteria. According to the “Experimental design”, fishes were infected with 100 uL of this solution, meaning 10^8 total bacteria. This is the real amount of pathogen used for infection and what should be indicated rather than the concentration without specifying the actual amount of pathogen

Thank you. As mentioned in abstract line 29,

“Group 1 was vaccinated intraperitoneal with 100 uL of FKVh on week 0 and booster dose was similarly administered on week 2. Group 2 was similarly injected with PBS. Skin mucus, serum and gut lavage were collected weekly for enzyme-linked immunosorbent assay (ELISA) and lysozyme activity assay from a total of 30 fish of each group. On week 4, the remaining 60 fish of each of Groups 1 and 2 were challenged with 108 cfu/fish of live Vibrio harveyi. “

8

How can the authors know the final concentration of the vaccine? For collecting the bacteria previous to their killing, they centrifuge the culture at 12.000 rpm. However, after fixation, the washing were performed by centrifugation at 2000 rpm. What is the reason for this change? Are fixed bacteria pelleted at this low speed?

Thank you for the typing error comments. Sorry for that. I already did a correction in preparation of FKVh vaccine as mentioned in line 103, page 3;

“Then, the bacterial cells were subsequently washed with PBS by centrifugation at 12,000 rpm and re-suspended”

9

The vaccine safety test were performed by injecting 1mL of the vaccine (I guess that at 10^9 CFUs), therefore injecting 10^9 FKVh). Was this so high dose performed on purpose?

To test for the vaccine safety, usually we double the volume of the vaccine or the dose must be higher than the dose that we want to use in our study. This study according to the previous research done by Pulpitat et al., 2020.

If the vaccine is safe enough then no changes happened to the host after vaccination was given. In this study, vaccine dose was 106 cfu/fish and the dose used for vaccine safety test is 109 cfu/fish

10

In order to make informative the histological images, please, show pictures from both vaccinated and non-vaccinated individual to compare the histological changes.

The author already added two histological images from group 1 (non-vaccinated group). Refer to line 321, page 10

C

C

C

(e) Brain tissue from vaccinated fish showing mild congestion (C) (H&E x100)

(f) Gill from a vaccinated fish showing the present of gill-associated lymphoid tissue (GIALT) (H&E x400)

Reviewer 2 Report

The author study formalin-killed V. harveyi strain Vh1 provided a better protection against vibriosis in marine red hybrid tilapia model by producing higher antibody responses and more efficient innate defence. Therefore, this FKVh vaccine is a suitable vaccine candidate against vibriosis.
  1. The introduction need re-write, the author could reference the new paper in cell titled why and how vaccines works. And cite the paper titled Identification and screening of effective protective antigens for channel catfish against Streptococcus iniae. Oncotarget. 2017 May 9; 8(19): 30793–30804. PMID: 28415641
  2. In Abstract :
vaccination is recommended to control this disease. disease

Vaccination is a recommended method to control fish diseases (Costa et al. 2011)

Here should be described as below.

Vaccination is a potential approach for prevention and control of disease in fish.
  1. In the manuscript, some parts the font size is strange
Such as:  34. Chatterjee, S., & Haldar, S. (2012). Vibrio related diseases in aquaculture and 465 development of rapid and accurate identification methods. Journal of Marine Science 466 Research and Development S, 1(1), 1-7.

Acknowledgments: The study was financially supported by the Higher Institution Centre of 370 Excellence, HICOE
  1. Li, J., Ma, S, and Woo, N.Y.S. (2016). Vaccination of silver sea bream (Sparus 417 sarba) against Vibrio alginolyticus: Protective evaluation of different vaccinating 418 modalities. International Journal of Molecular Science, 17(1), 40

  1. In part 4. Pathological Changes in Organs   Vaccinated group figures must provid.
  1. In Materials and Methods Preparation of Formalin-Killed Vibrio Harveyi (FKVh)  Vaccine Vaccine Safety Test  Experimental Design in those parts all need references.
  2. In Figure 3. Lysozyme activity a) mucus, b) serum and c) gut lavage of marine red hybrid tilapia following vaccination with formalin-killed harveyi for both Group 1 and 2. We require the author provid the figure use dot-plot represents.
(In summary, a Dot Plot is a graph for displaying the distribution of numerical variables where each dot represents a value. For whole numbers, if a value occurs more than once, the dots are placed one above the other so that the height of the column of dots represents the frequency for that value.)

Author Response

Revised version: 10 November 2020

Submission ID: vaccines-985551

Title: EFFICACY OF WHOLE CELL INACTIVATED VIBRIO HARVEYI VACCINE AGAINST VIBRIOSIS IN MARINE RED HYBRID TILAPIA (Oreochromis niloticus × O. mossambicus) MODEL

Reviewer 2

1

The author study formalin-killed V. harveyi strain Vh1 provided a better protection against vibriosis in marine red hybrid tilapia model by producing higher antibody responses and more efficient innate defence. Therefore, this FKVh vaccine is a suitable vaccine candidate against vibriosis.

Thank you.

2

The introduction need re-write, the author could reference the new paper in cell titled why and how vaccines works. And cite the paper titled Identification and screening of effective protective antigens for channel catfish against Streptococcus iniae. Oncotarget. 2017 May 9; 8(19): 30793–30804. PMID: 28415641

Thank you. Additional information as described in the manuscript. Please refer line 58, page 2.

“Extracellular and surface proteins of bacterial pathogens have the advantage of easy recognition by the infected host and thus are more likely to serve as targets for vaccine development. Actually, an abundance of extracellular or surface proteins of fish bacteria pathogens has been found to possess vaccine potential and confer immune protection (Wang et al.,2017).”

3

In Abstract:

vaccination is recommended to control this disease. disease

Vaccination is a recommended method to control fish diseases (Costa et al. 2011)

Here should be described as below.

Vaccination is a potential approach for prevention and control of disease in fish.

Thank you. Additional information as described in the manuscript in abstract line 22.

“Vibrio harveyi causes vibriosis in various commercial marine fish species. The infection leads to significant economic losses for aquaculture farms, and vaccination an alternative approach for prevention and control this disease in fish”

4

In the manuscript, some parts the font size is strange;

Such as:  34. Chatterjee, S., & Haldar, S. (2012). Vibrio related diseases in aquaculture and 465 development of rapid and accurate identification methods. Journal of Marine Science 466 Research and Development S, 1(1), 1-7.

Acknowledgments: The study was financially supported by the Higher Institution Centre of Excellence, HICOE

  1. Li, J., Ma, S, and Woo, N.Y.S. (2016). Vaccination of silver sea bream (Sparus 417 sarba) against Vibrio alginolyticus: Protective evaluation of different vaccinating 418 modalities. International Journal of Molecular Science, 17(1), 40

  1. In part 4. Pathological Changes in Organs   Vaccinated group figures must provid.
  1. In Materials and Methods Preparation of Formalin-Killed Vibrio Harveyi (FKVh)  Vaccine Vaccine Safety Test  Experimental Design in those parts all need references.
  2. In Figure 3. Lysozyme activity a) mucus, b) serum and c) gut lavage of marine red hybrid tilapia following vaccination with formalin-killed harveyi for both Group 1 and 2. We require the author provid the figure use dot-plot represents.

Thank you, all corrections have been improved accordingly.

5

In summary, a Dot Plot is a graph for displaying the distribution of numerical variables where each dot represents a value. For whole numbers, if a value occurs more than once, the dots are placed one above the other so that the height of the column of dots represents the frequency for that value.

The author decide to choose bar chart to present lysozyme activity as for the author, the info and data was clearly showed and easier to understand. Some previous studies also presented their lysozyme results using bar chart such as from study done by

“Harikrishnan, R., Balasundaram, C., & Heo, M. S. (2009). Effect of chemotherapy, vaccines and immunostimulants on innate immunity of goldfish infected with Aeromonas hydrophilaDiseases of aquatic organisms, 88(1), 45-54.”

End of respond

Round 2

Reviewer 1 Report

These are my comments to the author`s reply. Thank you for the effort.

1.- OK

2.- OK

3.- OK

4.- Thank you for the explanation. Now that I understand that the main goal of this work is not the vaccine formulation per se, but the use of hybrid tilapia as a vaccination model to replace more expensive fishes, I think this should be stated in the abstract. It is true that the last paragraph in the introduction explain this fact, but for non-expert readers, it is important to clarify this point also in the abstract.

5.- Thank you for the clarification. Still, figure 4 shows that the concentration used is not LD50 but around LD80. Thus, you might consider to reformulate this sentence or to include de standard deviation for this LD50 according to the indicated preliminary study.

6.- As the authors agree with my suggestion, this point should be discussed, introducing the provided references.

7.- OK.

8.- OK.

9.- Thank you for the clarification. A brief sentence stating this notion could be helpful.

10.- Thank you. Please, homogenate the labeling of the figures. Some of them have not the magnification or the legend letter. In addition, the magnification of the same tissue between vaccinated and non-vaccinated animals should be comparable. This is not the case for brain. Is figure 5c from vaccinated or non-vaccinated group?. It is not indicated in the figure legend.

Author Response

Revised version: 18 November 2020

Submission ID: vaccines-985551

Title: EFFICACY OF WHOLE CELL INACTIVATED VIBRIO HARVEYI VACCINE AGAINST VIBRIOSIS IN MARINE RED HYBRID TILAPIA (Oreochromis niloticus × O. mossambicus) MODEL

Reviewer 1

1

Thank you for the explanation. Now that I understand that the main goal of this work is not the vaccine formulation per se, but the use of hybrid tilapia as a vaccination model to replace more expensive fishes, I think this should be stated in the abstract. It is true that the last paragraph in the introduction explain this fact, but for non-expert readers, it is important to clarify this point also in the abstract.

Thank you. The explanation already inserted in abstract in Line 26

“Tilapia are fast growth, cheaper in price, resistant to disease and tolerance in adverse environmental condition from fresh water, brackish and marine water and because of this advantages, marine red hybrid tilapia is a suitable candidate as a fish model in fish disease and fish vaccination study especially in vibriosis by studying the immune response and histopathological changes after live bacteria was introduced.”

2

Thank you for the clarification. Still, figure 4 shows that the concentration used is not LD50 but around LD80. Thus, you might consider to reformulate this sentence or to include de standard deviation for this LD50 according to the indicated preliminary study.

Thank you. The sentence already reformulated on Page 3

3

As the authors agree with my suggestion, this point should be discussed, introducing the provided references

Thank you. The explanation already added in introduction Line 308.

“Formalin-killed vaccine are whole cell inactivated vaccine and this types of vaccine was choosen because from previous study, they stated that there was insignificant difference between both types of vaccine in mortality, survival rates’s results and effectiveness between both (Dehghani et al., 2012; Bactol et al., 2018). “

4

Thank you for the clarification. A brief sentence stating this notion could be helpful.

Thank you. A little explanation already inserted in Line 117.

If the vaccine is safe enough then no changes happened to the host after vaccination was given. In this study, vaccine dose was 106 cfu/fish and the dose used for vaccine safety test is 109 cfu/fish

5

Thank you. Please, homogenate the labeling of the figures. Some of them have not the magnification or the legend letter. In addition, the magnification of the same tissue between vaccinated and non-vaccinated animals should be comparable. This is not the case for brain. Is figure 5c from vaccinated or non-vaccinated group?. It is not indicated in the figure legend.

Thank you. Figure 5 already improvised based on comments

Reviewer 2 Report

 previous concerned questions not answered (no dot plot data , we don't know the data are true or not.)

You should change the figure to the same as the attachment

Author Response

Revised version: 18 November 2020

Submission ID: vaccines-985551

Title: EFFICACY OF WHOLE CELL INACTIVATED VIBRIO HARVEYI VACCINE AGAINST VIBRIOSIS IN MARINE RED HYBRID TILAPIA (Oreochromis niloticus × O. mossambicus) MODEL

Reviewer 2

1

Previous concerned questions not answered (no dot plot data, we don't know the data are true or not.)

You should change the figure to the same as the attachment

Thank you. The figure has been corrected based on comments from the bar chart to dot plot in Figure 3 (a), (b) and (c).

Round 3

Reviewer 1 Report

All my concerns have been addressed.

Just two minor comments.

In line 314, change "both" for "diverse". Non of the other forms of vaccines have been commented before this sentence, so it is not indicated the use of "both".

Panel 5e shows two (e) overlapped. Please correct.

Author Response

Revised version: 20 November 2020

Submission ID: vaccines-985551

Title: EFFICACY OF WHOLE CELL INACTIVATED VIBRIO HARVEYI VACCINE AGAINST VIBRIOSIS IN MARINE RED HYBRID TILAPIA (Oreochromis niloticus × O. mossambicus) MODEL

Reviewer 1

1

In line 314, change "both" for "diverse". Non of the other forms of vaccines have been commented before this sentence, so it is not indicated the use of "both".

Thank you. The word ‘both’ already changed to ‘adverse’ in Line 314.

“Formalin-killed vaccine, a whole cell inactivated vaccine was chosen in this study since previous studies had concluded that there was insignificant difference between adverse types of vaccine in term of mortality and rate of survival (Dehghani et al., 2012; Bactol et al., 2018).”

2

Panel 5e shows two (e) overlapped. Please correct.

Thank you. Correction already made in Figure 5 (e)

Reviewer 2 Report

Now make sense!

Author Response

Revised version: 20 November 2020

Submission ID: vaccines-985551

Title: EFFICACY OF WHOLE CELL INACTIVATED VIBRIO HARVEYI VACCINE AGAINST VIBRIOSIS IN MARINE RED HYBRID TILAPIA (Oreochromis niloticus × O. mossambicus) MODEL

Reviewer 2

1

Now make sense!

Thank you so much